# A Control-Centric Benchmark for Video Prediction

**Stephen Tian, Chelsea Finn, & Jiajun Wu**
Stanford University

## Abstract

Video is a promising source of knowledge for embodied agents to learn models of the world's dynamics. Large deep networks have become increasingly effective at modeling complex video data in a self-supervised manner, as evaluated by metrics based on human perceptual similarity or pixel-wise comparison. However, it remains unclear whether current metrics are accurate indicators of performance on downstream tasks. We find empirically that for planning robotic manipulation, existing metrics can be unreliable at predicting execution success. To address this, we propose a benchmark for action-conditioned video prediction in the form of a control benchmark that evaluates a given model for simulated robotic manipulation through sampling-based planning. Our benchmark, Video Prediction for Visual Planning ($VP^2$), includes simulated environments with 11 task categories and 310 task instance definitions, a full planning implementation, and training datasets containing scripted interaction trajectories for each task category. A central design goal of our benchmark is to expose a simple interface – a single forward prediction call – so it is straightforward to evaluate almost any action-conditioned video prediction model. We then leverage our benchmark to study the effects of scaling model size, quantity of training data, and model ensembling by analyzing five highly-performant video prediction models, finding that while scale can improve perceptual quality when modelling visually diverse settings, other attributes such as uncertainty awareness can also aid planning performance. Additional environment and evaluation visualizations are at this link.

## 1 Introduction

Dynamics models can empower embodied agents to solve a range of tasks by enabling downstream policy learning or planning. Such models can be learned from many types of data, but video is one modality that is task-agnostic, widely available, and enables learning from raw agent observations in a self-supervised manner. Learning a dynamics model from video can be formulated as a *video prediction* problem, where the goal is to infer the distribution of future video frames given one or more initial frames as well as the actions taken by an agent in the scene. This problem is challenging, but scaling up deep models has shown promise in domains including simulated games and driving (Oh et al., 2015; Harvey et al., 2022), as well as robotic manipulation and locomotion (Denton & Fergus, 2018; Villegas et al., 2019; Yan et al., 2021; Babaeizadeh et al., 2021; Voleti et al., 2022).

As increasingly large and sophisticated video prediction models continue to be introduced, how can researchers and practitioners determine which ones to use in particular situations? This question remains largely unanswered. Currently, models are first trained on video datasets widely adopted by the community (Ionescu et al., 2014; Geiger et al., 2013; Dasari et al., 2019) and then evaluated on held-out test sets using a variety of perceptual metrics. Those include metrics developed for image and video comparisons (Wang et al., 2004), as well as recently introduced deep perceptual metrics (Zhang et al., 2018; Unterthiner et al., 2018). However, it is an open question whether perceptual metrics are predictive of other qualities, such as planning abilities for an embodied agent.

In this work, we take a step towards answering this question for one specific situation: how can we compare action-conditioned video prediction models in downstream robotic control? We propose a benchmark for video prediction that is centered around robotic manipulation performance. Our benchmark, which we call the **Video Prediction for Visual Planning Benchmark** ($VP^2$), evaluates predictive models on manipulation planning performance by standardizing all elements of a control

setup *except* the video predictor. It includes simulated environments, specific start/goal task instance specifications, training datasets of noisy expert video interaction data, and a fully configured model-based control algorithm.

For control, our benchmark uses visual foresight (Finn & Levine, 2017; Ebert et al., 2018), a model-predictive control method previously applied to robotic manipulation. Visual foresight performs planning towards a specified goal by leveraging a video prediction model to simulate candidate action sequences and then scoring them based on the similarity between their predicted futures and the goal. After optimizing with respect to the score (Rubinstein, 1999; de Boer et al., 2005; Williams et al., 2016), the best action sequence is executed for a single step, and replanning is performed at each step. This is a natural choice for our benchmark for two reasons: first, it is goal-directed, enabling a single model to be evaluated on many tasks, and second, it interfaces with models only by calling forward prediction, which avoids prescribing any particular model class or architecture.

The main contribution of this work is a set of benchmark environments, training datasets, and control algorithms to isolate and evaluate the effects of prediction models on simulated robotic manipulation performance. Specifically, we include two simulated multi-task robotic manipulation settings with a total of 310 task instance definitions, datasets containing 5000 noisy expert demonstration trajectories for each of 11 tasks, and a modular and lightweight implementation of visual foresight.

Through our experiments, we find that models that score well on frequently used metrics can suffer when used in the context of control, as shown in Figure 1. Then, to explore how we can develop better models for control, we leverage our benchmark to analyze other questions such as the effects of model size, data quantity, and modeling uncertainty. We empirically test recent video prediction models, including recurrent variational models as well as a diffusion modeling approach. We will open source the code and environments in the benchmark in an easy-to-use interface, in hopes that it will help drive research in video prediction for downstream control applications.

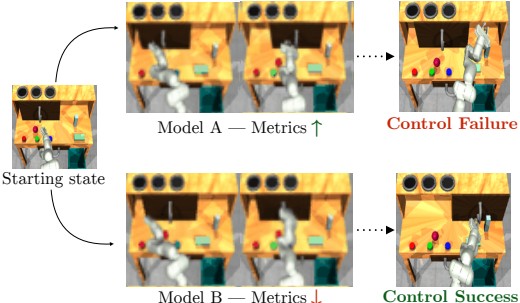

Figure 1: Models that score well on perceptual metrics may generate crisp but physically infeasible predictions that lead to planning failures. Here, Model A predicts that that the slide will move on its own.

## 2 RELATED WORK

**Evaluating video prediction models.** Numerous evaluation procedures have been proposed for video prediction models. One widely adopted approach is to train models on standardized datasets (Geiger et al., 2013; Ionescu et al., 2014; Srivastava et al., 2015; Cordts et al., 2016; Finn et al., 2016; Dasari et al., 2019) and then compare predictions to ground truth samples across several metrics on a held-out test set. These metrics include several image metrics adapted to the video case, such as the widely used $\ell_2$ per-pixel Euclidean distance and Peak signal-to-noise ratio (PSNR). Image metrics developed to correlate more specifically with human perceptual judgments include structural similarity (SSIM) (Wang et al., 2004), as well as recent methods based on features learned by deep neural networks like LPIPS (Zhang et al., 2018) and FID (Heusel et al., 2017). FVD (Unterthiner et al., 2018) extends FID to the video domain via a pre-trained 3D convolutional network. While these metrics have been shown to correlate well with human perception, it is not clear whether they are indicative of performance on control tasks. Geng et al. (2022) develop correspondence-wise prediction losses, which use optical flow estimates to make losses robust to positional errors. These losses may improve control performance and are an orthogonal direction to our benchmark.

Another class of evaluation methods judges a model's ability to make predictions about the outcomes of particular physical events, such as whether objects will collide or fall over (Sanborn et al., 2013; Battaglia et al., 2013; Bear et al., 2021). This excludes potentially extraneous information from the rest of the frame. Our benchmark similarly measures only task-relevant components of predicted videos, but does so through the lens of overall control success rather than hand-specified questions. Oh et al. (2015) evaluate action-conditioned video prediction models on Atari games by training a Q-learning agent using predicted data. We evaluate learned models for planning rather than policy learning, and extend our evaluation to robotic manipulation domains.

| Model | Loss | Perceptual | | | Control |
| --- | --- | --- | --- | --- | --- |
| | | FVD | LPIPS* | SSIM | Success |
| FitVid | MSE | 30.7 | 3.4 | 87.8 | 65% |
| | +LPIPS=1 | **18.0** | **2.8** | **89.3** | 67% |
| | +LPIPS=10 | 24.3 | 4.1 | 84.6 | 35% |
| SVG′ | MSE | 51.7 | 5.1 | 82.7 | **80%** |
| | +LPIPS=1 | 40.7 | 4.4 | 83.2 | **80%** |
| | +LPIPS=10 | 45.1 | 4.8 | 81.8 | 37% |

(a) `robosuite` pushing tasks

| Model | Loss | Perceptual | | | Control |
| --- | --- | --- | --- | --- | --- |
| | | FVD | LPIPS* | SSIM | Success |
| FitVid | MSE | 9.0 | **0.62** | 97.4 | 58% |
| | +LPIPS=1 | **5.9** | 0.63 | **97.5** | **82%** |
| | +LPIPS=10 | 6.8 | 0.70 | 97.3 | 32% |
| SVG′ | MSE | 10.6 | 0.97 | 95.3 | 70% |
| | +LPIPS=1 | 7.2 | 0.89 | 95.5 | 73% |
| | +LPIPS=10 | 24.2 | 1.1 | 94.0 | 10% |

(b) RoboDesk: push red button

| Model | Loss | Perceptual | | | Control |
| --- | --- | --- | --- | --- | --- |
| | | FVD | LPIPS* | SSIM | Success |
| FitVid | MSE | 20.5 | **1.25** | **94.4** | 50% |
| | +LPIPS=1 | 9.8 | 1.26 | 93.3 | 75% |
| | +LPIPS=10 | **7.3** | 1.30 | 92.8 | **83%** |
| SVG′ | MSE | 18.3 | 1.68 | 91.3 | 47% |
| | +LPIPS=1 | 11.0 | 1.58 | 90.9 | 68% |
| | +LPIPS=10 | 18.9 | 1.76 | 90.4 | 20% |

(c) RoboDesk: upright block off table

| Model | Loss | Perceptual | | | Control |
| --- | --- | --- | --- | --- | --- |
| | | FVD | LPIPS* | SSIM | Success |
| FitVid | MSE | 15.1 | **1.08** | **95.8** | 38% |
| | +LPIPS=1 | 10.2 | **1.08** | 94.9 | 36% |
| | +LPIPS=10 | **9.8** | 1.39 | 93.6 | 13% |
| SVG′ | MSE | 22.5 | 1.88 | 90.6 | **58%** |
| | +LPIPS=1 | 4.9 | 2.06 | 89.7 | 10% |
| | +LPIPS=10 | 22.6 | 2.48 | 88.2 | 10% |

(d) RoboDesk: open slide

Table 1: Perceptual metrics and control performance for models trained using a MSE objective, as well as with added perceptual losses. For each metric, the bolded number shows the best value for that task. *LPIPS scores are scaled by 100 for convenient display. Full results can be found in Appendix G.

**Benchmarks for model-based and offline RL.** Many works in model-based reinforcement learning evaluate on simulated RL benchmarks (Brockman et al., 2016; Tassa et al., 2018; Ha & Schmidhuber, 2018; Rajeswaran et al., 2018; Yu et al., 2019; Ahn et al., 2019; Zhu et al., 2020; Kannan et al., 2021), while real-world evaluation setups are often unstandardized. Offline RL and imitation learning benchmarks (Zhu et al., 2020; Fu et al., 2020; Gulcehre et al., 2020; Lu et al., 2022) provide training datasets along with environments. Our benchmark includes environments based on the infrastructure of `robosuite` (Zhu et al., 2020) and RoboDesk (Kannan et al., 2021), but it further includes task specifications in the form of goal images, cost functions for planning, as well as implementations of planning algorithms. Additionally, offline RL benchmarks mostly analyze model-free algorithms, while in this paper we focus on model-based methods. Because planning using video prediction models is sensitive to details such as control frequency, planning horizon, and cost function, our benchmark supplies all aspects other than the predictive model itself.

## 3 THE MISMATCH BETWEEN PERCEPTUAL METRICS AND CONTROL

In this section, we present a case study that analyzes whether existing metrics for video prediction are indicative of performance on downstream control tasks. We focus on two variational video prediction models that have competitive prediction performance and are fast enough for planning: FitVid (Babaeizadeh et al., 2021) and the modified version of the SVG model (Denton & Fergus, 2018) introduced by Villegas et al. (2019), which contains convolutional as opposed to fully-connected LSTM cells and uses the first four blocks of VGG19 as the encoder/decoder architecture. We denote this model as SVG′. We perform experiments on two tabletop manipulation environments, `robosuite` and RoboDesk, which each admit multiple potential downstream task goals. Additional environment details are in Section 4.

When selecting models to analyze, our goal is to train models that have varying performance on existing metrics. One strategy for learning models that align better with human perceptions of realism is to add auxiliary perceptual losses such as LPIPS (Zhang et al., 2018). Thus, for each environment, we train three variants of both the FitVid and SVG′ video prediction models. One variant is trained with a standard pixel-wise $\ell_2$ reconstruction loss (MSE), while the other two are trained using an additional perceptual loss in the form of adding the LPIPS score with VGG features implemented by Kastryulin et al. (2022) between the predicted and ground truth images at weightings 1 and 10. We train each model for 150K gradient steps. We then evaluate each model in terms of FVD (Unterthiner et al., 2018), LPIPS (Zhang et al., 2018), and SSIM (Wang et al., 2004) on held-out validation sets, as well as planning performance on robotic manipulation via visual foresight (Finn & Levine, 2017;

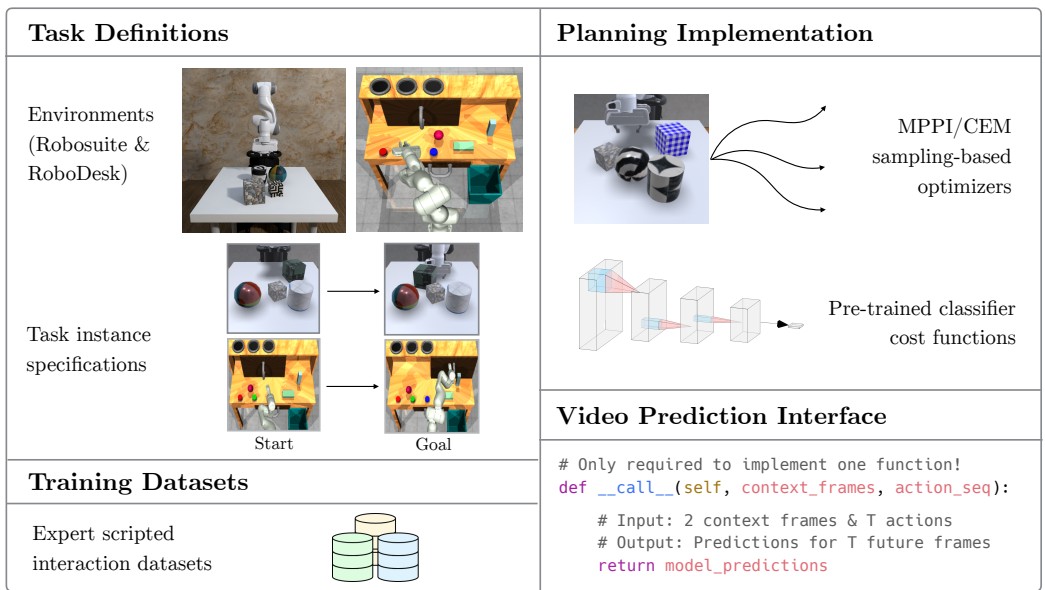

Figure 2: Overview of the VP² benchmark, which contains simulated environments, training data, and task instance specifications, along with a planning implementation with pre-trained cost functions. The interface for evaluating a new model on the benchmark is a model forward pass.

Ebert et al., 2018). As shown in Table 1, we find that models that show improved performance on these metrics do not always perform well when used for planning, and the degree to which they are correlated with control performance appears highly task-dependent.

For example, FVD tends to be low for models that are trained with an auxiliary LPIPS loss at weight 10 on `robosuite` pushing, despite weak control performance. At the same time, for the `upright block off table` task, FVD is much better correlated with task success compared to LPIPS and SSIM, which show little relation to control performance. We also see that models can perform almost identically on certain metrics while ranging widely in control performance.

This case study is conducted in a relatively narrow setting, and because data from these simulated environments is less complex than "in-the-wild" datasets, the trained models tend to perform well on an absolute scale across all metrics. Existing metrics can certainly serve as effective indicators of general prediction quality, but we can observe even in this example that they can be misaligned with a model's performance when used for control and could lead to erroneous model selection for downstream manipulation. Therefore, we believe that a control-centric benchmark represents another useful axis of comparison for video prediction models.

## 4  THE VP² BENCHMARK

In this section, we introduce the VP² benchmark. The main goal of our benchmark is to evaluate the downstream control performances of video prediction models. To help isolate the effects of models as opposed to the environment or control algorithm, we include the entire control scheme for each task as part of the benchmark. We design VP² with three intentions in mind: (1) it should be *accessible*, that is, evaluating models should not require any experience in control or RL, (2) it should be *flexible*, placing as few restrictions as possible on models, and (3) it should *emphasize settings where model-based methods may be advantageous*, such as specifying goals on the fly after training on a multi-task dataset.

VP² consists of three main components: **environment and task definitions**, **a sampling-based planner**, and **training datasets**. These components are shown in Figure 2.

### 4.1  ENVIRONMENT AND TASK DEFINITIONS

One advantage of model-based methods is that a single model can be leveraged to complete a variety of commanded goals at test time. Therefore, to allow our benchmark to evaluate models'

abilities to perform multi-task planning, we select environments where there are many different tasks for the robot to complete. VP$^2$ consists of two *environments*: a tabletop setting created using `robosuite` (Zhu et al., 2020), and a desk manipulation setting based on RoboDesk (Kannan et al., 2021). Each environment contains a robot model as well as multiple objects to interact with.

For each environment, we further define *task categories*, which represent semantic tasks in that environment. Each task category is defined by a success criterion based on the simulator state.

Many reinforcement learning benchmarks focus on a single task and reset the environment to the same simulator state at the beginning of every episode. This causes the agent to visit a narrow distribution of states, and does not accurately represent how robots operate in the real world. However, simply randomizing initial environment states can lead to higher variance in evaluation. To test models' generalization capabilities and robustness to environment variations, we additionally define *task instances* for each task category. A *task instance* is defined by an initial simulator state and an RGB goal image $I_g \in \mathbb{R}^{64 \times 64 \times 3}$ observation of the desired final state. Goal images are a flexible way of specifying goals, but can often be unachievable based on the initial state and action space. We ensure that for each task instance, it is possible to reach an observation matching the goal image within the time horizon by collecting them from actual trajectories executed in the environment.

VP$^2$ currently supports evaluation on 11 tasks across two simulated environments:

**Tabletop `robosuite` environment.** The tabletop environment is built using `robosuite` (Zhu et al., 2020) and contains 4 objects: two cubes of varying sizes, a cylinder, and a sphere. To provide more realistic visuals, we render the scene with the iGibson renderer (Li et al., 2021). This environment contains 4 task categories: push {`large cube`, `small cube`, `cylinder`, `sphere`}. We include 25 task instances per category, where the textures of all objects are randomized from a set of 13 for each instance. When selecting the action space, we prefer a lower-dimensional action space to minimize the complexity of sampling-based planning, but would like one that can still enable a range of tasks. We choose to use an action space $\mathcal{A} \in \mathbb{R}^4$ that fixes the end-effector orientation and represents a change in end-effector position command, as well as an action opening or closing the gripper, that is fed to an operational space controller (OSC).

**RoboDesk environment.** The RoboDesk environment (Kannan et al., 2021) consists of a Franka Panda robot placed in front of a desk, and defines several tasks that the robot can complete. Because the planner needs to be able to judge task success from several frames, we consider a subset of these tasks that have obvious visual cues for success. Specifically, we use 7 tasks: push {`red`, `green`, `blue`} `button`, open {`slide`, `drawer`}, push {`upright block`, `flat block`} `off table`. We find that the action space in the original RoboDesk environment works well for scripting data collection and for planning. Therefore in this environment $\mathcal{A} \in \mathbb{R}^5$ represents a change in gripper position, wrist angle, and gripper command.

## 4.2 SAMPLING-BASED PLANNING

Our benchmark provides not only environment and task definitions, but also a control setup that allows models to be directly scored based on control performance. Each benchmark run consists of a series of control trials, where each control trial executes sampling-based planning using the given model on a particular task instance. At the end of $T$ control steps, the success or failure on the task instance is judged based on the simulator state.

To perform planning using visual foresight, at each step the sampling-based planner attempts to solve the following optimization problem to plan an action sequence given a goal image $I_g$, context frames $I_c$, cost function $C$, and a video prediction model $\hat{f}_\theta$: $\min_{a_1, a_2, \ldots a_T} \sum_{i=1}^{T} C(\hat{f}(I_c, a_{1:T})_i, I_g)$. The best action is then selected, and re-planning is performed at each step to reduce the effect of compounding model errors. We use 2 context frames and predict $T = 10$ future frames across the benchmark. As in prior work in model-based RL, we implement a sampling-based planner that uses MPPI (Williams et al., 2016; Nagabandi et al., 2019) to sample candidate action sequences, perform forward prediction, and then update the sampling distribution based on these scores. We provide default values for planning hyperparameters that have been tuned to achieve strong performance with a perfect dynamics model, which can be found along with additional details in Appendix B.

VP$^2$ additionally specifies the cost function $C$ for each task category. For the task categories in the `robosuite` environment, we simply use pixel-wise mean squared error (MSE) as the cost. For

the RoboDesk task categories, we find that an additional task-specific pretrained classifier yields improved planning performance. We train deep convolutional networks to classify task success on each task, and use a weighted combination of MSE and classifier logits as the cost function. We provide these pre-trained model weights as part of the benchmark.

### 4.3  Training datasets

Each environment in $VP^2$ comes with datasets for video prediction model training. Each training dataset consists of trajectories with 35 timesteps, each containing $256 \times 256$ RGB image observations and the action taken at each step. Specifics for each environment dataset are as follows, with additional details in Appendix D:

- **`robosuite` Tabletop environment**: We include 50K trajectories of interactions collected with a hand-scripted policy to push a random object in the environment in a random direction. Object textures are randomized in each trajectory.
- **RoboDesk environment**: For each task instance, we include 5K trajectories collected with a hand-scripted policy, for a total of 35K trajectories. To encourage the dataset to contain trajectories of varying success rates, we apply independent Gaussian noise to each dimension of every action from the scripted policy before executing it.

## 5  Benchmark Interface

One of our goals is to make the benchmark easy as possible to use, without placing restrictions on the deep learning framework nor requiring expertise in RL or planning. We achieve this through two main design decisions. First, by our selection of a sampling-based planning method, we remove as many assumptions as possible from the model definition, such as differentiability or an autoregressive predictive structure. Second, by implementing and abstracting away control components, we establish a code interface that requires minimal overhead.

To evaluate a model on $VP^2$, models must first be trained on one dataset per environment. This is an identical procedure to typical evaluation on video benchmarking datasets. Then, with a forward pass implementation, our benchmark uses the model directly for planning as described in Section 4.2. Specifically, given context frames $[I_1, I_2, ..., I_t]$ and an action sequence $[a_1, a_2, ..., a_{t+T-1}]$, the forward pass should predict the next $N$ frames $[\hat{I}_{t+1}, \hat{I}_{t+2}, ..., \hat{I}_{t+T}]$. We anticipate this will incur low overhead, as similar functions are often implemented to track model validation performance.

While this interface is designed for for ease-of-use and comparison, $VP^2$ can also be used in an "open" evaluation format where controllers may be modified, to benchmark entire planning systems.

## 6  Empirical Analysis of Video Prediction at Scale for Control

Next, we leverage our benchmark as a starting point for investigating questions relating to model and data scale in the context of control. We first evaluate a set of baseline models on our benchmark tasks. Then in order to better understand *how* we can build models with better downstream control performance on $VP^2$, we empirically study the following questions:

- How does control performance scale with model size? Do different models scale similarly? What are the computational costs at training and test time?
- How does control performance scale with training data quantity?
- Can planning performance be improved by models with better uncertainty awareness that can detect when they are queried on out-of-distribution action sequences?

### 6.1  Performance of Existing Models on $VP^2$

To establish performance baselines, we consider five models that achieve either state-of-the-art or competitive performance on metrics such as SSIM, LPIPS, and FVD. See Appendix A for details.

- **FitVid** (Babaeizadeh et al., 2021) is a variational video prediction model that achieved state-of-the-art results on the Human 3.6M (Ionescu et al., 2014) and RoboNet (Dasari et al., 2019) datasets. It has shown the ability to fit large diverse datasets, where previous models suffered from underfitting.

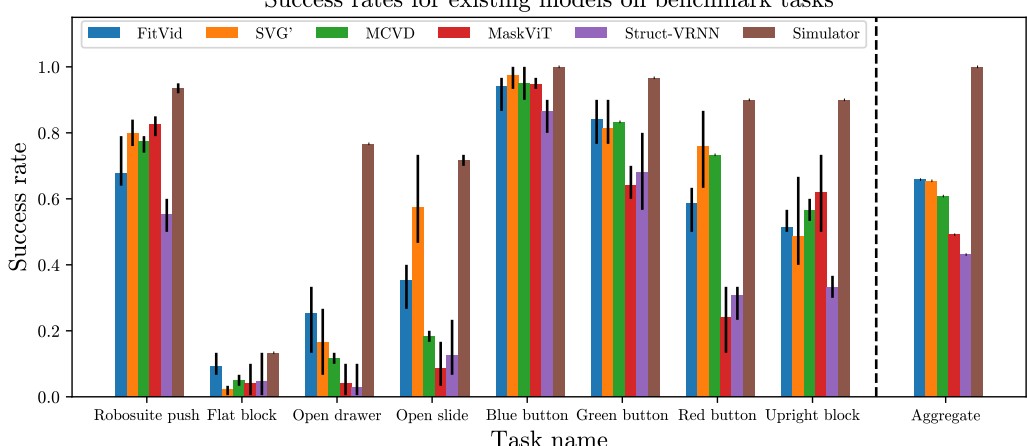

Figure 3: Performance of existing models on VP$^2$. We aggregate results over the 4 `robosuite` task categories. Error bars show min/max performance over 5 control runs, except for MCVD where we use 3 runs due to computational constraints. On the right, we show the mean scores for each model averaged across all tasks, normalized by the performance of the simulator.

- **SVG′** (Villegas et al., 2019) is also a variational RNN-based model, but makes several unique architectural choices – it opts for convolutional LSTMs and a shallower encoder/decoder. We use the $\ell_2$ loss as in the original SVG model rather than $\ell_1$ prescribed by Villegas et al. (2019) to isolate the effects of model architecture and loss function.

- **Masked Conditional Video Diffusion (MCVD)** (Voleti et al., 2022) is a diffusion model that can perform many video tasks by conditioning on different subsets of video frames. To our knowledge, diffusion-based video prediction models have not previously been applied to learn action-conditioned models. We adapt MCVD to make it action-conditioned by following a tile-and-concatenate procedure similar to Finn et al. (2016).

- **Struct-VRNN** (Minderer et al., 2019) uses a keypoint-based representation for dynamics learning. We train Struct-VRNN without keypoint sparsity or temporal correlation losses for simplicity, finding that they do not significantly impact performance on our datasets.

- **MaskViT** (Gupta et al., 2022) uses a masked prediction objective and iterative decoding scheme to enable flexible and efficient inference using a transformer-based architecture.

We train each model except MCVD to predict 10 future frames given 2 context frames and agent actions. For MCVD, we also use a planning horizon of 10 steps, but following the procedure from the paper, we predict 5 future frames at a time, and autoregressively predict longer sequences.

While we can compare relative planning performance between models based on task success rate, it is difficult to evaluate *absolute* performance when models are embedded into a planning framework. To provide an upper bound on how much the dynamics model can improve control, we include a baseline that uses the simulator directly as the dynamics but retains the planning pipeline. This disentangles weaknesses of video prediction models from suboptimal planning or cost functions.

In Figure 3, we show the performance of the five models on our benchmark tasks. We see that for the simpler task `push blue button`, the performance of existing models approaches that of the true dynamics. However, for `robosuite` and the other RoboDesk tasks, there are significant gaps in performance for learned models.

## 6.2 Model Capacity

Increasingly expressive models have pushed prediction quality on visually complex datasets. However, it is unclear how model capacity impacts downstream manipulation results on VP$^2$ tasks. In this section, we use our benchmark to study this question. Due to computational constraints, we consider only tasks in the `robosuite` tabletop environment. We train variants of the FitVid and SVG′ models with varying parameter counts. For FitVid, we create two smaller variants by halving the number of encoder layers and decreasing layer sizes in one model, and then further decreasing the number of encoder filters and LSTM hidden units in another ("mini"). For SVG′, we vary the

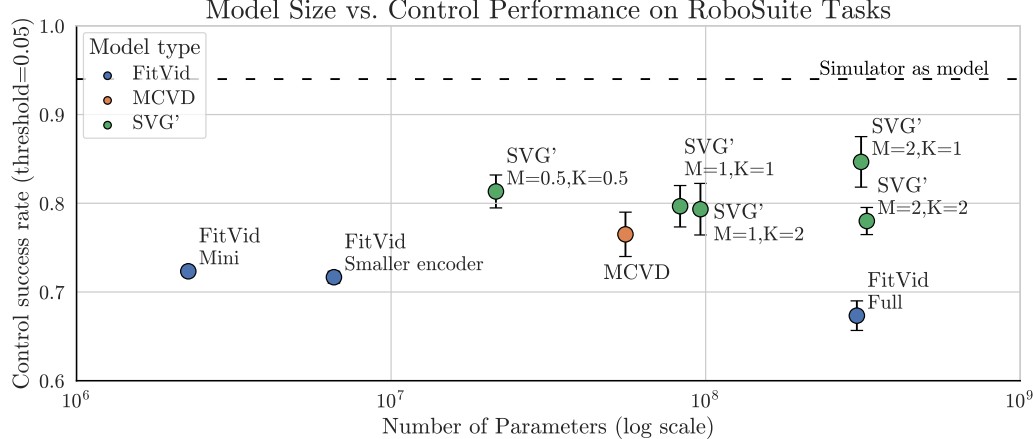

Figure 4: Control performance on `robosuite` tasks across a models with capacities ranging from 6 million to 300 million parameters. Error bars represent standard error of the mean (SEM) across 3 control runs over the set of 100 task instances, except for MCVD where we perform 2 control runs due to computational constraints. We hypothesize that larger versions of FitVid overfit the data, and see that in general, model capacity does not seem to yield signficantly improved performance on these tasks.

model size by changing parameters $M$ and $K$ as described by Villegas et al. (2019), which represent expanding factors on the size of the LSTM and encoder/decoder architectures respectively.

In Figure 4, we plot the control performance on the `robosuite` tabletop environment versus the number of parameters. While SVG′ sees slight performance improvements with certain larger configurations, in general we do not see a strong trend that increased model capacity yields improved performance. We hypothesize that this is because larger models like the full FitVid architecture tend to overfit to action sequences seen in the dataset.

We additionally note the wall clock forward prediction time for each model in Table 2. Notably, while the MCVD model achieves competitive control performance, its forward pass computation time is more than $10\times$ that of the full FitVid model. While diffusion models have

| Model | Variant | # Params | Pred. time (s) |
|---|---|---|---|
| FitVid | Full | 302M | 5.63 |
| | Small encoder | 6.5M | 0.48 |
| | Mini | 2.3M | 0.29 |
| SVG′ | $M = 2, K = 2$ | 325M | 3.58 |
| | $M = 2, K = 1$ | 312M | 2.40 |
| | $M = 1, K = 2$ | 96M | 2.39 |
| | $M = 1, K = 1$ | 83M | 1.21 |
| | $M = \frac{1}{2}, K = \frac{1}{2}$ | 21.5M | 0.52 |
| MCVD | Base | 56M | 220 |

Table 2: Comparison of median wall clock forward pass time for predicting 10 future frames. We use a batch size of 200 samples and one NVIDIA Titan RTX GPU.

shown comparable prediction quality compared to RNN-based video prediction models, the challenge of using these models efficiently for planning remains.

## 6.3 DATA QUANTITY

Data-driven video prediction models require sufficient coverage of the state space to be able to perform downstream tasks, and much effort has gone into developing models that are able to fit large datasets. Therefore, it is natural to ask: How does the number of training samples impact downstream control performance? To test this, we train the full FitVid and base SVG′ models on subsets of the `robosuite` tabletop environment dataset consisting of 1K, 5K, 15K, 30K, and 50K trajectories of 35 steps each. We then evaluate the control performance for each model for the aggregated 100 `robosuite` control tasks. Figure 5 shows the control results for models trained on consecutively increasing data quantities. We find that performance improves when increasing from smaller quantities of data on the order of hundreds to thousands of trajec-

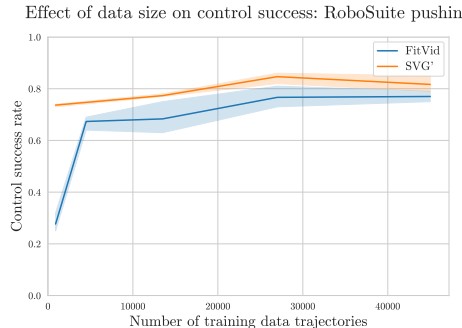

Figure 5: Evaluating the effects of increasing training dataset size on all `robosuite` tasks. Additional data boosts performance slightly, but the benefit quickly plateaus. Shaded areas show 95% confidence intervals across 3 runs.

tories, but gains quickly plateau. We hypothesize that this is because the distribution of actions in the dataset is relatively constrained due to the scripted data collection policy.

### 6.4 UNCERTAINTY ESTIMATION VIA ENSEMBLES

Because video prediction models are trained on fixed datasets before being deployed for control in visual foresight, they may yield incorrect predictions when queried on out-of-distribution actions during planning. As a result, we hypothesize that planning performance might be improved through improved uncertainty awareness – that is, a model should detect when it is being asked to make predictions it will likely make mistakes on. We test this hypothesis by estimating epistemic uncertainty through a simple ensemble disagreement method and integrating it into the planning procedure.

Concretely, we apply a score penalty during planning based on the instantiation of ensemble disagreement from Yu et al. (2020). Given an ensemble of $N = 4$ video prediction models, we use all models from the ensemble to perform prediction for each action sequence. Then, we compute the standard task score using a randomly selected model, and then subtract a penalty based on the largest $\ell_1$ model deviation from the mean prediction. Because subtracting the score penalty decreases the scale of rewards, we experiment with the temperature $\gamma = 0.01, 0.03, 0.05$ for MPPI, and report the best control result for each task across values of $\gamma$ for single and ensembled models. Additional details about the penalty computation can be found in Appendix E.

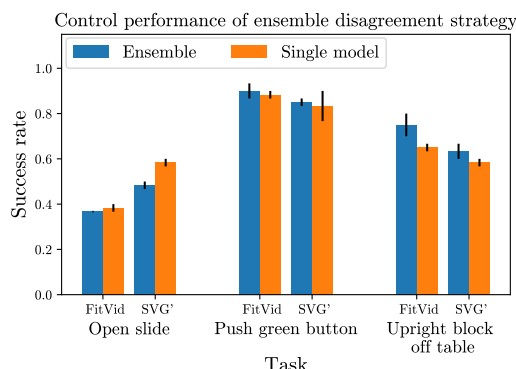

Figure 6: Control performance when using ensemble disagreement for control on three tasks. We can see that for `upright block off table`, ensemble disagreement improves performance, but on the other two tasks, performance is comparable or slightly weaker than a single model. Error bars show min/max performance across 2 control runs.

The results of using this approach for control are shown in Figure 6. We find that the uncertainty penalty is able to achieve slightly improved performance on the `upright block off table` task, comparable performance on `push green button`, and slightly weaker performance on `open slide`. This also causes expensive training and inference times to scale linearly with ensemble size when using a naïve implementation. However, our results indicate that efforts in developing uncertainty-aware models may be one avenue for improved downstream planning performance.

## 7 CONCLUSION

In this paper, we proposed a control-centric benchmark for video prediction. After finding empirically that existing perceptual and pixel-wise metrics can be poorly correlated with downstream performance for planning robotic manipulation, we proposed VP$^2$ as an additional *control-centric* evaluation method. Our benchmark consists of 13 task categories in two simulated multi-task robotic manipulation environments, and has an easy-to-use interface that does not place any limits on model structure or implementation. We then leveraged our benchmark to investigate questions related to model and data scale for five existing models – and find that while scale is important to some degree for VP$^2$ tasks, improved performance on these tasks may also come from building models with improved uncertainty awareness. We hope that this spurs future efforts in developing action-conditioned video prediction models for downstream control applications, and also provides a helpful testbed for creating new planning algorithms and cost functions for visual MPC.

**Limitations.** VP$^2$ consists of simulated, shorter horizon tasks. As models and planning methods improve, it may be extended to include more challenging tasks. Additionally, compared to widely adopted metrics, evaluation scores on our benchmark are more computationally intensive to obtain and may be less practical to track over the course of training. Finally, robotic manipulation is one of many downstream tasks for video prediction, and our benchmark may not be representative of performance on other tasks. We anticipate evaluation scores on VP$^2$ to be used in conjunction with other metrics for a holistic understanding of model performance.

REPRODUCIBILITY STATEMENT

We provide our open-sourced for the benchmark in Appendix F. We have also released the training datasets and pre-trained cost function weights at that link.

ACKNOWLEDGMENTS

We thank Tony Zhao for providing the initial FitVid model code, Josiah Wong for guidance with `robosuite`, Fei Xia for help customizing the iGibson renderer, as well as Yunzhi Zhang, Agrim Gupta, Roberto Martín-Martín, Kyle Hsu, and Alexander Khazatsky for helpful discussions. This work is in part supported by ONR MURI N00014-22-1-2740 and the Stanford Institute for Human-Centered AI (HAI). Stephen Tian is supported by an NSF Graduate Research Fellowship under Grant No. DGE-1656518.

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

## A  MODEL TRAINING DETAILS

In this section, we provide specific training details for video prediction model training here.

### A.1  FITVID

We reimplement FitVid according to the official implementation by the authors. We train using the architecture defaults from the original paper. The training hyperparameters are shown in Table 3. For the smaller variants shown in the model capacity experiments, we detail the modifications in architecture in Table 4 and 5.

We train FitVid at $64 \times 64$ image resolution to predict 10 future frames given 2 future frames.

When training, we use FP16 precision using PyTorch's autocast functionality and use either 2 or 4 NVIDIA TITAN RTX GPUs. We train all models for 153K gradient steps.

| Hyperparameter | Value |
|---|---|
| Batch size | 32 |
| Optimizer | Adam |
| Learning rate | 3e-4 |
| Adam $\epsilon$ | 1e-8 |
| Gradient clip | 100 |
| $\beta$ | 1e-4 |

Table 3: Hyperparameters for FitVid training. Note that we use a learning rate of $3e-4$ because we find that it allows for more stable training. We also tried $1e-3$ as in the original paper, but we found that this tended to cause numerical instability and did not yield significantly different results.

| Hyperparameter | Value |
|---|---|
| Encoder (h) dimension | 64 |
| LSTM size | 128 |
| Num of encoder/decoder layers per stage | $[1, 1, 1, 1]$ |

Table 4: Hyperparameters for FitVid model: smaller encoder.

| Hyperparameter | Value |
|---|---|
| Encoder (h) dimension | 32 |
| LSTM size | 64 |
| Num of encoder/decoder layers per stage | $[1, 1, 1, 1]$ |
| Encoder/Decoder # filters | All scaled by 1/4 |

Table 5: Hyperparameters for FitVid model: mini.

### A.2  SVG$'$

For SVG$'$, we start with the implementation of the SVG model by Denton & Fergus (2018). Then, we make the modifications described by Villegas et al. (2019). Specifically, we use the first 4 blocks of VGG as the encoder/decoder architecture. Then, rather than flattening encoder outputs before feeding them into the LSTM layers, we use Convolutional LSTMs to directly process the 2D feature maps. Unless otherwise stated, we train with the model size $M = 1, K = 1$, which is the base model size described in Villegas et al. (2019). Note that for fairer comparisons, unlike Villegas et al. (2019), we retain the $\ell_2$ loss for the reconstruction portion of the loss function.

For action conditioning with SVG$'$, we tile the action for each timestep into a 2D feature map with the same dimensions as the encoded image, where the number of channels is the action size. We then concatenate this along with the latent $z$ to the encoded image before passing it into the frame predictor RNN.

When testing higher and lower capacity variants, we adjust the values of $M$ and $K$, which as defined by Villegas et al. (2019), are hyperparameters that multiplicatively scale the number of filters in the LSTM layers and encoder/decoder, respectively.

We train SVG$'$ at $64{\times}64$ image resolution to predict 10 future frames given 2 context frames. We use 1-2 NVIDIA TITAN RTX GPUs to train SVG$'$ for 153K gradient steps. Training hyperparameters are shown in Table 6.

| Hyperparameter | Value |
|---|---|
| Batch size | 32 |
| Optimizer | Adam |
| Learning rate | 3e-4 |
| Adam $\beta_1$ | 0.9 |
| Adam $\beta_2$ | 0.999 |
| Adam $\epsilon$ | 1e-8 |

Table 6: Hyperparameters for SVG$'$ training.

### A.3 MCVD

We use the code implementation for MCVD provided by the original authors at `https://github.com/voletiv/mcvd-pytorch`. We use the SPATIN version of the model, following the setup for training the Denoising Diffusion Probabilistic Models (DDPM) version of the model on SMMNIST in the original paper.

We additionally modify the architecture to be action-conditioned by concatenating first converting a sequence of actions into a flat vector containing actions from all timesteps. We then tile this action vector $a \in \mathbb{R}^{T*|\mathcal{A}|}$ into a 2D feature map with the same spatial dimensions as the context images, and concatenate the feature map and context images channel-wise. Additional training hyperparameters are shown in Table 7.

| Hyperparameter | Value |
|---|---|
| Batch size | 64 |
| Optimizer | Adam |
| Learning rate | 2e-4 |
| Adam $\beta_1$ | 0.9 |
| Adam $\beta_2$ | 0.999 |
| Adam $\epsilon$ | 1e-8 |
| Gradient clip | 1.0 |
| Weight decay | 0.0 |
| # diffusion steps (train) | 100 |
| # diffusion steps (test) | 100 |

Table 7: Hyperparameters for MCVD training.

We train MCVD to generate $64 \times 64$ RGB images. MCVD can be used for a number of different video tasks based on which frames are provided for conditioning, but we use it for video prediction (only future frame prediction provided context). Although during planning we use MCVD to predict to a horizon of length 10 like other models, we train the model to predict 5 future frames given 2 context frames, following the procedure by the original authors. In order to make predictions of length 10, we feed in the results of the first prediction autoregressively to the model to obtain the last 5 predictions. Note that only the actions for the first 5 frames are provided for the first forward pass, and only the actions for the second 5 frames are provided in the second pass.

We train the MCVD model for `robosuite` tasks for 360K gradient steps, and the one for RoboDesk tasks for 270K gradient steps. Each model is trained on 2 NVIDIA TITAN RTX GPUs.

### A.4 MASKVIT

We use the hyperparameters used by the authors when training on the BAIR dataset, but with a slightly increased positional embedding size (1024 rather than 768). We did not tune these parameters further.

A.5 STRUCT-VRNN

We reimplement the Struct-VRNN model from Minderer et al. (2019). We tune the weighting of the KL-divergence parameter in the set of values $1e-0, 1e-1, 1e-2, 1e-3, 1e-4$, and use the value of $1e-4$, that achieves the best control performance on the RoboSuite tasks, for the rest of the experiments. We use 64 keypoints in the representation, matching the largest number used for any dataset in the original paper. Full hyperparameters are shown in Table 8.

| Hyperparameter | Value |
|---|---|
| Batch size | 4 |
| Optimizer | Adam |
| Learning rate | 3e-4 |
| Reconstruction loss weight | 10.0 |
| KL loss weight | 0.001 |
| Coordinate prediction loss | 1.0 |
| # keypoints | 64 |
| Keypoint $\sigma$ | 0.1 |
| Encoder initial number of filters | 32 |
| Appearance encoder initial number of filters | 32 |
| Decoder initial # of filters | 256 |
| Dynamics model hidden size | 512 |
| Prior/posterior # of layers | 2 |
| Prior/posterior hidden size | 512 |

Table 8: Hyperparameters for Struct-VRNN training.

# B PLANNING IMPLEMENTATION DETAILS

In this section we provide details for the planning implementation of VP$^2$. For all tasks, we use $T = 15$ as the rollout length.

## B.1 MPPI OPTIMIZER

We perform sampling-based planning using the model-predictive path integral (MPPI) (Williams et al., 2016), using the implementation from Nagabandi et al. (2019) as a guideline. Table 9 details the hyperparameters that we use for planning for each task category.

| Hyperparameter | Value |
|---|---|
| Number of samples | 200 for all tasks except 800 for `open {drawer, slide}` |
| Scaling factor $\gamma$ | 0.05 |
| Sampling distribution correlation coefficient $\beta$ | 0.5 |
| Sampling distribution stdev. (RoboDesk) | [0.5,0.5,0.5,0.1,0.1] |
| Sample distribution stdev. (`robosuite`) | [0.5, 0.5, 0.5, 0] |
| Sampling distribution initial mean | 0 |

Table 9: Hyperparameters for MPPI optimizer.

## B.2 CLASSIFIER COST FUNCTIONS

For each classifier cost function, we take 2500 trajectories from the RoboDesk dataset for the given task, and train a binary convolutional classifier to predict whether or not a given frame receives reward $>= 30$, where the reward is provided by the RoboDesk environment. We train for 3000 gradient steps, and the architecture is described in Table 10.

## B.3 PLANNING COST FUNCTIONS

Next we detail the cost functions for each of the tasks. The cost for the RoboSuite tasks is the sum of the $\ell_2$ pixel-wise error between the 10 predicted images and the goal image, summed over time. For the RoboDesk tasks, it is 0.5 times the $\ell_2$ pixel loss plus 10 times the classifier logit

| Layer type | Out channels/hidden units | Kernel |
|---|---|---|
| Conv2D | 32 | $3 \times 3$ |
| Conv2D | 32 | $3 \times 3$ |
| ReLU | - | - |
| Conv2D | 32 | $3 \times 3$ |
| ReLU | - | - |
| Conv2D | 32 | $3 \times 3$ |
| ReLU | - | - |
| Conv2D | 32 | $3 \times 3$ |
| ReLU | - | - |
| Conv2D | 32 | $3 \times 3$ |
| ReLU | - | - |
| Conv2D | 32 | $3 \times 3$ |
| ReLU | - | - |
| Conv2D | 32 | $3 \times 3$ |
| Flatten | - | - |
| Linear | 1024 | - |
| ReLU | - | - |
| Linear | 1 | - |

Table 10: Classifier cost function architecture, with layers named as in PyTorch convention. The input is a $64 \times 64 \times 3$ RGB image.

from a deep convolutional classifier trained for predicting success on that given task. We tuned this weighting value after fixing the value of $\gamma$ for MPPI. We tuned over the classifier weight as $[1, 10, 100, 1000, 10000]$ using the simulator as the planner and found that $10$ resulted in the best performance.

The score for the planner is computed by negating the cost.

## C  ALTERNATIVE PLANNERS

While we implement and tune the sampling-based MPPI optimizer for accessibility and ease of use, our code framework also enables benchmark users to modify and swap out the controller and model independently. This allows for the coupled development of models and controllers on the task and task instance definitions provided in the benchmark. We provide the following optimizers/controllers with the released codebase. A star (*) indicates that the method requires gradient information.

- **Model-predictive path integral (MPPI) (Williams et al., 2016)**: This is our default planner as described in the main text. Our implementation is based off of that of Nagabandi et al. (2019).
- **Cross-entropy method (CEM) (de Boer et al., 2005)**: The cross-entropy method iteratively refines the sampling distribution of candidate distributions via importance sampling. In practice, this is implemented by computing the top percentile of elite samples and recomputing the mean and variance at each iteration. Our implementation is based off of that of Nagabandi et al. (2019).
- **Cross-entropy method with Gradient Descent* (CEM-GD) (Huang et al., 2021)**: This is a gradient-augmented version of CEM that refines individual CEM samples using gradient steps. It also significantly reduces the number of CEM samples after the first planning step for computational speed. We keep most of the PyTorch implementation by the original authors.
- **Limited-memory BFGS* (L-BFGS) (Liu & Nocedal, 1989)**: A popular quasi-Newton method for continuous nonlinear optimization. We use the implementation from PyTorch (Paszke et al., 2019).

We conducted initial control experiments with these planners using the SVG$'$ model on the `robosuite` task categories. We find that CEM-GD promisingly achieves an $87\%$ success rate averaged across 3 seeds, and L-BFGS achieves a $48\%$ success rate across 3 seeds after tuning the learning rate for the optimizer across $\{$1e-3, 1e-2, 5e-2, 7e-2$\}$.

## D  DATASET DETAILS

Here we provide details abut the datasets that come with VP$^2$. Note that our datasets can be re-rendered at any resolution.

- **robosuite Tabletop environment**: We collect 50K trajectories of interactions collected with a scripted policy. During each trajectory, a random one of the four possible objects is selected as the target for the push. Then, a random direction in $[0, \pi]$ on the plane is selected as the direction for the object to be pushed, and the target object position is set to 0.3 meters in this direction from the initial starting position. Then, we use a P-controller to first navigate the arm in position to push the object in the desired direction, and then to push it. At every step, we add independent Gaussian noise with $\sigma = 0.05$ to each dimension of the action except the last one, which represents the gripper action. Even if the object is not successfully pushed to the desired position using this policy, we still record the trajectory.

    The robosuite Tabletop dataset is rendered using the iGibson (Li et al., 2021) renderer, with modifications to the shadow computations to make the shadows softer and more realistic. We will supply the modified rendering code along with the environments in the released benchmark code.

- **RoboDesk environment**: We script policies to complete each of the 7 tasks. The structure of each policy depends on the task, and they are included in the provided code. We apply noise to each action by adding independent Gaussian noise to every dimension. We create the dataset for the entire RoboDesk dataset by collecting 2500 noisy scripted trajectories using a noise level of $\sigma = 0.1$, and then 2500 additional trajectories with $\sigma = 0.2$, for a total of 35K trajectories.

    The RoboDesk dataset is rendered using the MuJoCo viewer, provided by the original implementation.

## E    ENSEMBLING EXPERIMENTS

For the ensembling experiment, we apply the model disagreement penalty in the following way. Given an action sequence, we first use all $N$ models in the ensemble to predict video sequences $I^1, I^2, ...I^N$. We then compute the error of the prediction that most deviates from the mean of all predictions in $\ell_1$ error, i.e. $\delta = \max_{i=[0,1,...N]} \|I^i - \frac{1}{N}\sum_{i=1}^{N} I^i\|_1$. We compute the standard task cost function $c$ using one of the ensemble predictions, selected uniformly at random. We calculate the final cost as $c - \lambda\delta$, where $\lambda$ is a hyperparameter that we set as 0.01 for all experiments.

## F    CODE

We provide the open source code used for running control experiments here: `https://github.com/s-tian/vp2`. Pretrained cost weights, datasets, and task definitions can be downloaded from that link.

## G    FULL RESULTS FOR METRIC COMPARISONS

In Table 11 we present the results on the remaining two tasks from the case study presented in Section 3.

In Figures 7-14, we provide detailed per-task plots of LPIPS, FVD, and SSIM compared to control performance. We find that while none of these metrics is well-correlated with performance across all tasks, they appear to be better correlated in the "push blue button", "push green button", and "open drawer" tasks. However, we can see that even for those tasks, these metrics can conflict in ordering models. For specific tasks, we observe that the "upright block off table" task particularly appears well-correlated with FVD.

| Model | Loss | Perceptual | | | Control |
|---|---|---|---|---|---|
| | | FVD | LPIPS* | SSIM | Success |
| FitVid | MSE | 9.6 | **0.65** | **98.1** | 95% |
| | +LPIPS=1 | **6.3** | 0.72 | 98.0 | **98%** |
| | +LPIPS=10 | 9.2 | 0.88 | 97.8 | 88% |
| SVG′ | MSE | 16.7 | 1.13 | 95.3 | 97% |
| | +LPIPS=1 | 8.4 | 1.06 | 95.5 | 97% |
| | +LPIPS=10 | 41.8 | 1.28 | 94.1 | 25% |

(a) RoboDesk: push blue button

| Model | Loss | Perceptual | | | Control |
|---|---|---|---|---|---|
| | | FVD | LPIPS* | SSIM | Success |
| FitVid | MSE | 10.6 | **0.65** | **98.0** | 88% |
| | +LPIPS=1 | 8.3 | 0.69 | 97.9 | 88% |
| | +LPIPS=10 | 8.3 | 0.87 | 97.4 | 67% |
| SVG′ | MSE | 13.1 | 1.13 | 94.9 | 83% |
| | +LPIPS=1 | **7.4** | 1.03 | 95.3 | 83% |
| | +LPIPS=10 | 24.6 | 1.26 | 93.8 | 10% |

(b) RoboDesk: push green button

| Model | Loss | Perceptual | | | Control |
|---|---|---|---|---|---|
| | | FVD | LPIPS* | SSIM | Success |
| FitVid | MSE | 9.1 | 0.77 | **96.3** | 10% |
| | +LPIPS=1 | **5.8** | **0.72** | 96.0 | 10% |
| | +LPIPS=10 | 7.2 | 0.91 | 94.8 | 10% |
| SVG′ | MSE | 13.6 | 1.28 | 92.8 | 10% |
| | +LPIPS=1 | 9.7 | 1.19 | 92.7 | **13%** |
| | +LPIPS=10 | 20.0 | 1.54 | 91.0 | 10% |

(c) RoboDesk: flat block off table

Table 11: Results for the remaining 3 tasks for the experiment described in Section 3.

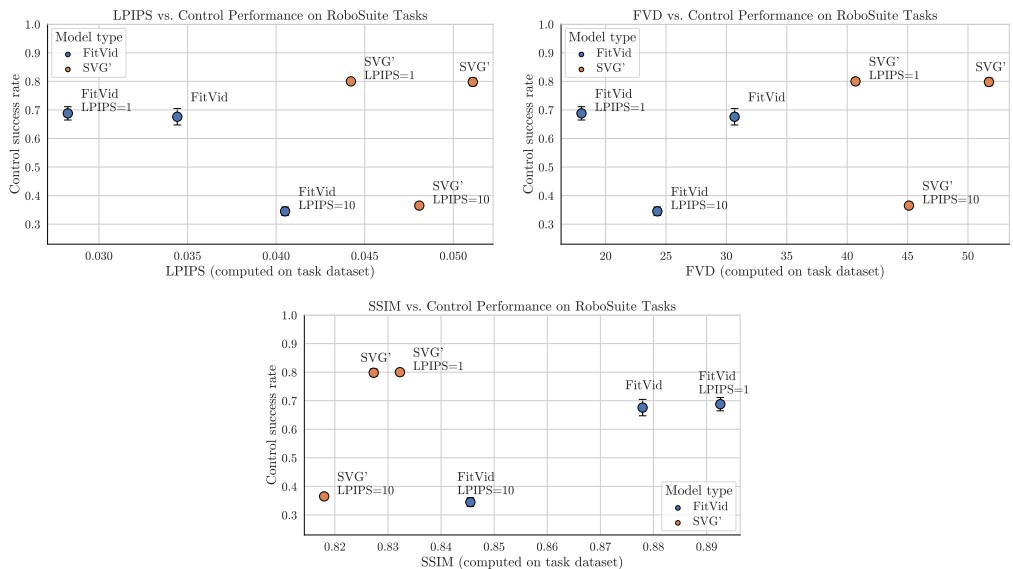

Figure 7: Detailed results comparing perceptual metric values and control performance: RoboSuite tasks.

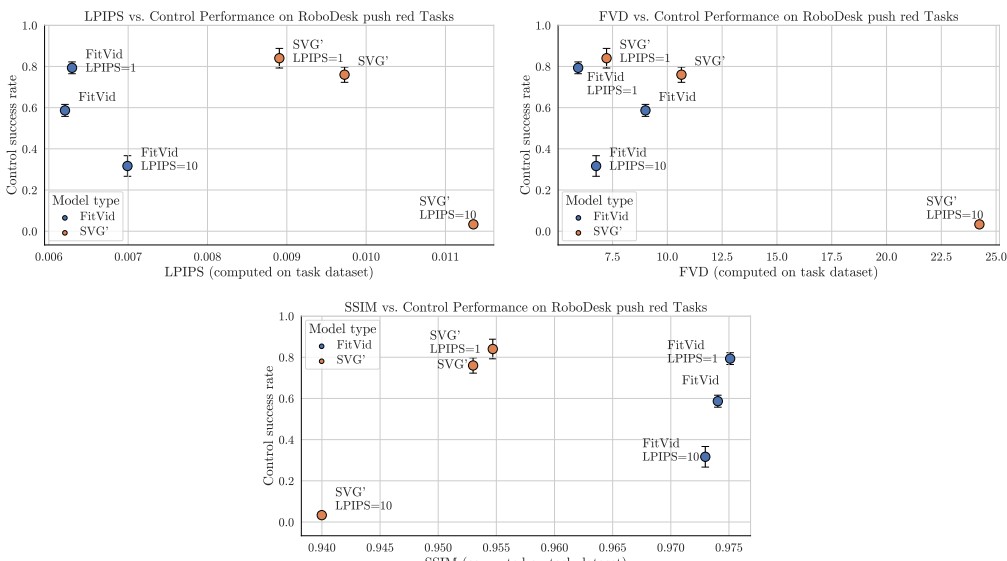

Figure 8: Detailed results comparing perceptual metric values and control performance: RoboDesk push red button task.

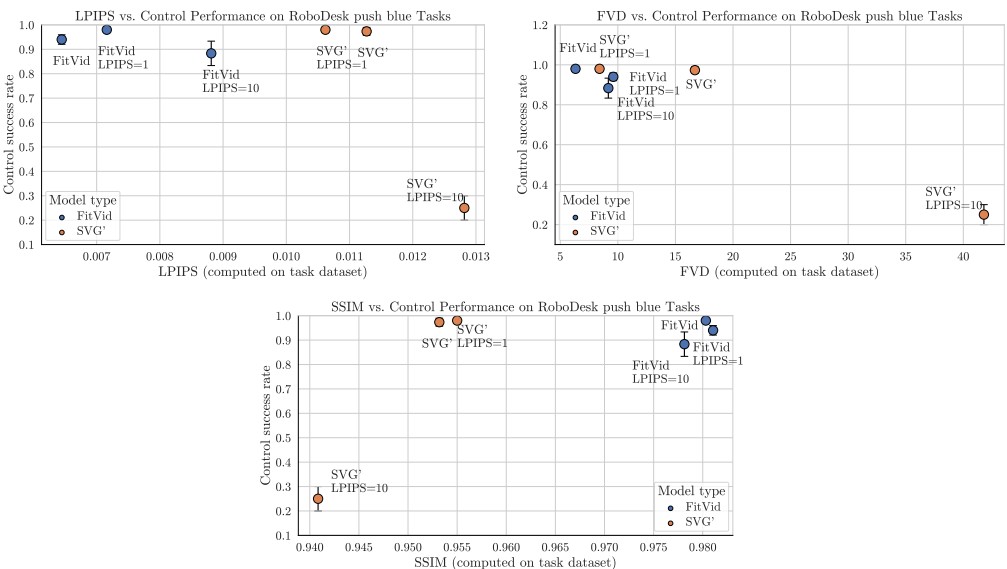

Figure 9: Detailed results comparing perceptual metric values and control performance: RoboDesk push blue button task.

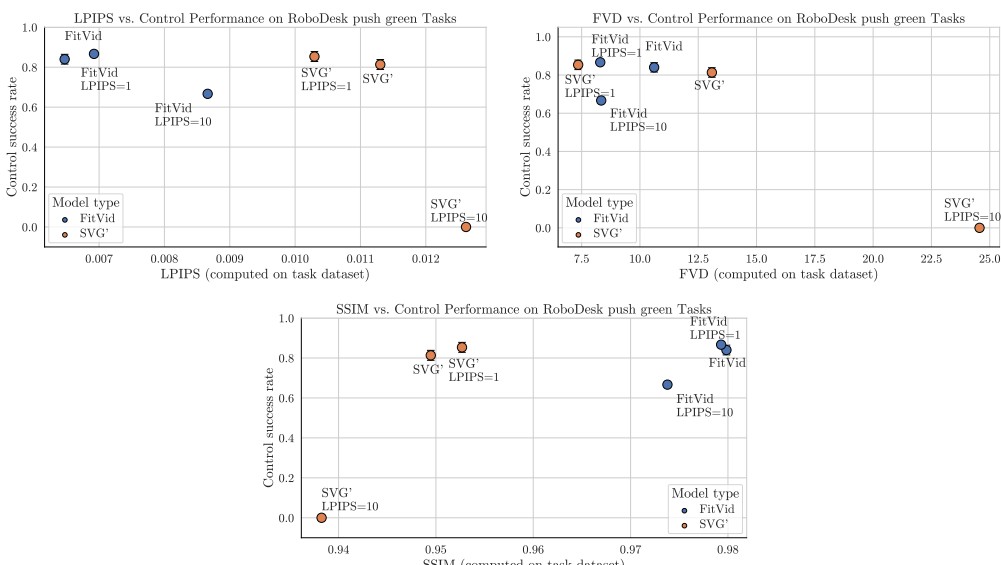

Figure 10: Detailed results comparing perceptual metric values and control performance: RoboDesk push green button task.

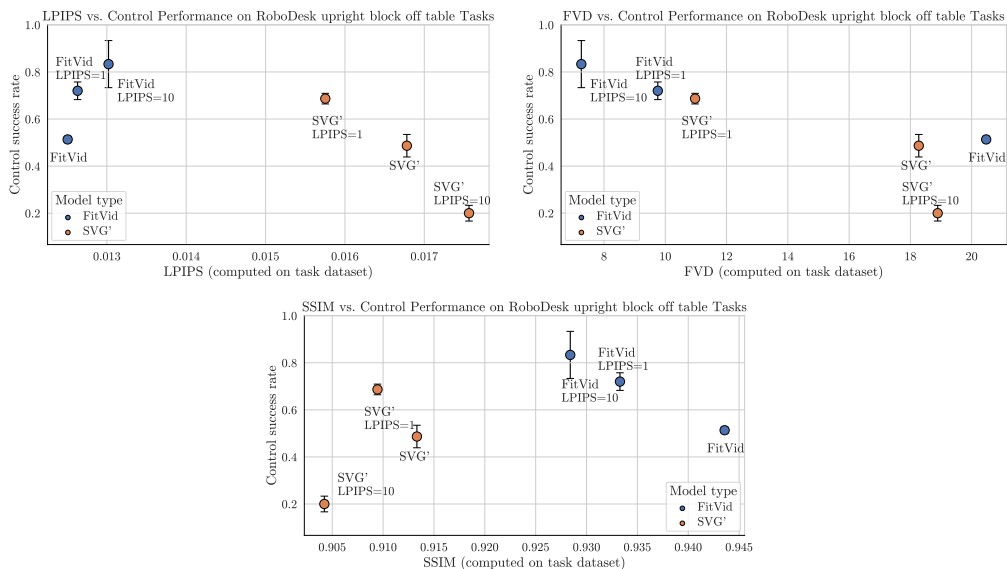

Figure 11: Detailed results comparing perceptual metric values and control performance: RoboDesk upright block off table task.

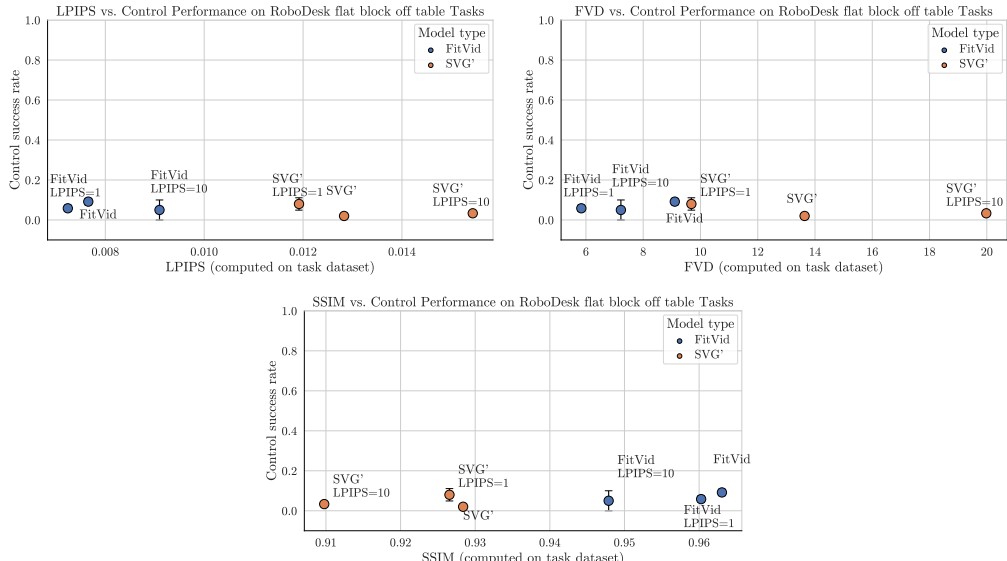

Figure 12: Detailed results comparing perceptual metric values and control performance: RoboDesk flat block off table task.

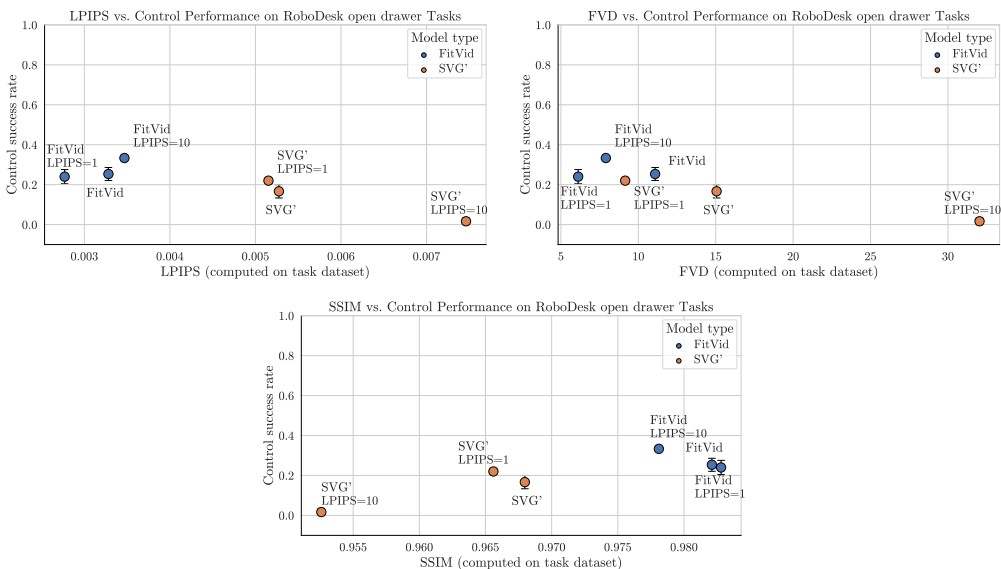

Figure 13: Detailed results comparing perceptual metric values and control performance: RoboDesk open drawer task.

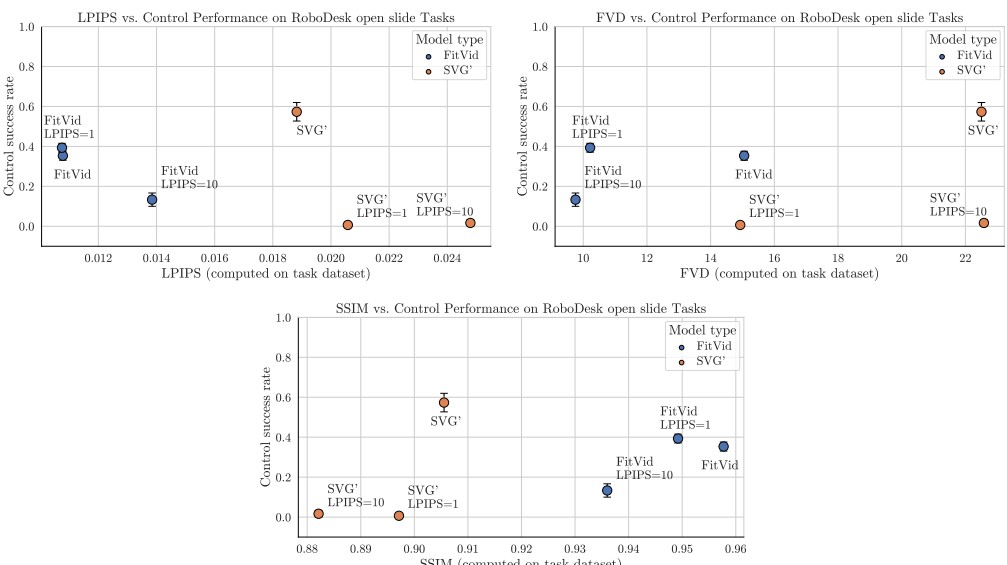

Figure 14: Detailed results comparing perceptual metric values and control performance: RoboDesk open slide task.

