# OpenReview forum: "A Control-Centric Benchmark for Video Prediction"
_ICLR.cc/2023/Conference — ICLR 2023 poster_

### Official Review · Reviewer_wekY · 2022-10-23

**Confidence:** 3
**Correctness:** 3
**Technical Novelty And Significance:** 2
**Empirical Novelty And Significance:** 3
**Recommendation:** 6

**Clarity, Quality, Novelty And Reproducibility:**

- The paper is well-motivated and easy to understand. The accompanying website provides a nice overview of the paper the benchmark.
- The paper definitely proposes a novel benchmark, which is also important to complement existing video prediction evaluation.
- The paper should be easy to reproduce since the authors promise to release the code soon.

**Strength And Weaknesses:**

**Strength:**

- The paper studies an important problem, i.e., the evaluation of generative models, which is hard since there are no predefined metrics. The paper provides an example of a better evaluation of generative video prediction models using downstream control tasks.
- The paper proposes a good case study that shows popular metrics such as LPIPS and MSE may not reflect the true performance of downstream control tasks, which justifies the need for the proposed method.
- The benchmark features a simple plug-and-play interface that makes it easy to use and general for almost all video prediction methods.

**Weakness:**

- As a benchmark paper, the number of baselines is too few. Only three methods are evaluated while there are many more video prediction methods that could be evaluated given the simple and general interface.
- The performance is reported with 2 runs only (in Figure 3), which might not be enough to accurately capture the performance and variance.
- Only a specific planning method with specific hyperparameters is used for evaluation, which might not be indicative, since different video prediction (dynamics) models may require different algorithms or hyperparameters to perform the best.
- The ensemble experiment is weak since it was only showcased for 3 tasks where the improvement is only significant for one task while the ensemble actually performs worse for another task.

**Summary Of The Paper:**

The paper presents a benchmark that evaluates video prediction models using the performance of downstream control tasks to complement traditional perceptual similarity or pixel-wise metrics. Specifically, the benchmark exposes a simple forward dynamics interface for the video prediction model to use it in a model-based planning method, visual foresight. The benchmark uses the robosuite tabletop environment and robodesk environment, where 11 robot manipulation tasks are designed in total. The paper evaluated three video prediction methods including variational RNN- and diffusion-based models.

**Summary Of The Review:**

This paper addresses an important problem, i.e., improving generative video prediction evaluation with downstream control tasks. It proposes an easy-to-use benchmark for this setting. While there are several concerns such as the number of baselines, I currently slightly lean toward acceptance.

---

### Official Review · Reviewer_5Vyr · 2022-10-23

**Confidence:** 4
**Correctness:** 3
**Technical Novelty And Significance:** 1
**Empirical Novelty And Significance:** 2
**Recommendation:** 6

**Clarity, Quality, Novelty And Reproducibility:**

The proposed benchmark is clearly described and motivated for. The benchmark extends existing settings that are relatively well established.

**Strength And Weaknesses:**

Strengths:

I think the empirical results showing that perceptual video predictive quality metrics do not correlate well with control success is an interesting finding that will likely be of value to others.

I agree that abstracting away control will result in significant performance increases in dynamics modelling and short term progress in this area.

Weaknesses:

The proposed benchmark abstracts controller details from the dynamics model in the name of simplicity and ease of use, but I am not convinced that this is a good decision when it comes to longer term progress in our field. I also think the benchmark itself is vulnerable to a number of the criticisms this paper makes about evaluating video prediction in isolation of the underlying control tasks. I have a strong suspicion that the same article could be written about this framework, with results showing that different dynamics models perform differently with different control strategies and planners (eg. CEM, MPPI, trajectory optimisation etc.) and arguing that we need a new benchmark to avoid this. It's likely that this framework will result in a set of good dynamics models for use with MPPI, but not with other approaches.

I am not convinced this will advance control performance, which may well benefit from more tight coupling between dynamics modelling and control rather than less. For this reason, I would be more inclined to use robosuite and robodesk directly, and value the ability to explore multiple control, planning and dynamics modelling approaches. I think this benchmark needs more controller options (beyond sampling based approaches) and to ideally enable the use of gradient based approaches that better leverage the learned dynamics models.

**Summary Of The Paper:**

This paper introduces a benchmark for evaluating vision-based dynamics models. The paper shows that common prediction metrics (typically revolving around predicted image quality and not physical plausibility or causal interaction effects) may not necessarily correlate with control performance, and thus proposes an evaluation approach structured around a number of tasks in the robodesk and robsuite environments. For this the controller is standardised (MPPI) and abstracted away to allow SOTA chasers to focus directly on the dynamics modelling aspects of the task.

**Summary Of The Review:**

I like the analysis of video prediction vs control, but am unconvinced that the proposed new benchmark is of significant value to advancing dynamics modelling for control. I think the benchmark needs more controller options (beyond sampling) and to allow gradient propagation to be really useful.

----- Post rebuttal updates ----

The authors have added additional controllers to the benchmark, addressing the primary concern I had with this work.

---

### Official Review · Reviewer_LW5N · 2022-10-25

**Confidence:** 4
**Correctness:** 3
**Technical Novelty And Significance:** 3
**Empirical Novelty And Significance:** 3
**Recommendation:** 8

**Clarity, Quality, Novelty And Reproducibility:**

The paper is generally clear and well-written. The benchmark and methods are well described and the experiments could be replicated. The authors provide the source code for the benchmark, therefore making sure that the community can use it.
The novelty of the paper is sort of secondary here: Although the authors are not the first to note that performance in robotic tasks is important for prediction, they provide, to my knowledge, the first benchmark of this sort. Therefore, I believe that this is a valuable contribution (despite my criticism earlier).

**Strength And Weaknesses:**

### Strengths
- The rationale for the paper is very good, and it is addressing a relevant question: How to find better metrics for video prediction (or image generation in general) is an important question for the field, and robotics tasks are a good choice.
- The availability of such a benchmark is likely to stimulate discussion and research in the video prediction field.
- The experiments are well-designed and thorough.
- The paper is generally clear and pleasant to read.

### Weaknesses
- The choice of tasks for the benchmark is of very limited variety. Out of the 11 tasks:
	- 4 are pushing objects of different (but convex) shapes
	- 3 are pushing buttons of different colour
	- 2 are pushing off blocks
	- and two are opening - admittedly the most interesting ones.
	This is a limitation as it is not clear that most of those tasks are really the best chosen to evaluate motion prediction in videos, and could very well fail to measure large errors in the prediction. More varied manipulation (stacking?) would be interesting.
- The number of methods evaluated in the benchmark is limited, although if the benchmark is provided to the community this is not a very crucial issue.
- The analysis of the result is limited, and provides only limited insights in the validity of the initial hypothesis. It would have been good, for example, to have a plot of pixel accuracy vs simulation accuracy of the different methods to assess whether the classical metrics are good proxy or not. Also, which tasks out of the 11 are correlated to which would also be an interesting consideration.

**Summary Of The Paper:**

This paper provides a new benchmark for evaluating video prediction approaches. The authors remark that the metrics used for evaluating video prediction quality, based on pixel accuracy, can be poor guides to the quality of the prediction. In some cases, an apparently accurate prediction may lead to failure in, eg, robotic planning. The authors propose to address this with a new benchmark evaluating video prediction from the perspective of 11 simulated robotic tasks, across two simulators.
State-of-the-art video predictions methods are compared using those tasks.

**Summary Of The Review:**

This is an interesting paper, that provides a useful and timely resource to the community to possibly help provide a broader understanding on the quality and the limitations of video prediction approaches. Despite my concerns that the chosen robotic tasks are too limited in variety, this is clearly a step in the right direction and would be very useful to the field.

---

### Official Review · Reviewer_fMXK · 2022-11-03

**Confidence:** 2
**Correctness:** 4
**Technical Novelty And Significance:** 3
**Empirical Novelty And Significance:** 3
**Recommendation:** 6

**Clarity, Quality, Novelty And Reproducibility:**

### Clarity

Very clear presentation.

### Quality

General good quality of presentation and general material.


### Novelty

The novelty consists on the benchmark with can have a very practical use.

### Reproducibility

The paper misses code, but if the authors provide code the benchmark should be fully reproducible.
The training details are well explained on supplementary material.

**Strength And Weaknesses:**


### Strengths

1. The paper guides the planning literature that maybe they shouldn't be using
reconstruction and perception losses when predicting future moves.
2. The paper is well written and the results are clear.


### Weaknesses

1. I feel the main weakness is the potential significance of the results
is limited as I feel people tend to know that reconstruction is not ideal.
I think the paper would be better if the authors actually found any
metric that seems to correlate more ?

**Summary Of The Paper:**

The papers presents a benchmark to evaluate if video prediction
quality metrics correlates with control/ robotics manipulation.
As expected good perceptual quality of prediction does not
necessarily indicates better control capabilities.

They compare diverse sota methods on the new benchmark
using different model sizes, data quantity etc.

**Summary Of The Review:**

My recommendation is for weak acceptance.
I think the benchmark is useful and give new insights about planning methods.
I am however, not so familiar with the literature and I might be wrong about
the significance of this proposal.

---

### Decision · Program_Chairs · 2023-01-20

**Decision:**

Accept: poster

**Justification For Why Not Higher Score:**

Limited insights and results from the given set of experiments.

**Justification For Why Not Lower Score:**

The problem domain is urgently needed along current research front, and would be of good amount of interest to ICLR audiences.

**Metareview: Summary, Strengths And Weaknesses:**

This work presents a new benchmark for evaluating video prediction approaches. The authors remark that the metrics used for evaluating video prediction quality, based on pixel accuracy, can be poor guides to the quality of the prediction. In some cases, an apparently accurate prediction may lead to failure in, eg, robotic planning. The paper evaluated three video prediction methods including variational RNN- and diffusion-based models.

+ The work studies a timely critical challenge, i.e., the evaluation of generative models.
+ The newly compiled benchmark features a simple plug-and-play interface that makes it easy to adopt;

All reviewers agree it is a decently presented benchmark paper, and the benchmark introduced has merit and could be further used. Meanwhile, the analysis of the result is limited, and provides only limited insights in the validity of the initial hypothesis.

Thus I would deem it at highest a poster paper.

**Note From Pc:**

if the above contains the word "oral" or "spotlight" please see: "oral" presentation means -> notable-top-5% and "spotlight" means -> notable-top-25%. As stated in our emails, we are disassociating presentation type from AC recommendations

**Summary Of Ac-Reviewer Meeting:**

N/A